ecology, biological applications

human–wildlife interaction, diet, phenology, population change, supplementary provisioning, breeding density

**Author for correspondence:**
Jack D. Shutt
e-mail: jackshutt8@gmail.com

# Faecal metabarcoding reveals pervasive long-distance impacts of garden bird feeding

Jack D. Shutt[1,2], Urmi H. Trivedi[3] and James A. Nicholls[2,4]

[1]Department of Natural Sciences, Manchester Metropolitan University, Manchester M1 5GD, UK
[2]Institute of Evolutionary Biology, University of Edinburgh, The King's Buildings, Edinburgh EH9 3FL, UK
[3]Edinburgh Genomics, University of Edinburgh, Edinburgh EH9 3FL, UK
[4]Australian National Insect Collection, CSIRO, Acton, Australian Capital Territory 2601, Australia

JDS, 0000-0002-4146-8748; JAN, 0000-0002-9325-563X

Supplementary feeding of wildlife is widespread, being undertaken by more than half of households in many countries. However, the impact that these supplemental resources have is unclear, with impacts largely considered to be restricted to urban ecosystems. We reveal the pervasiveness of supplementary foodstuffs in the diet of a wild bird using metabarcoding of blue tit (*Cyanistes caeruleus*) faeces collected in early spring from a 220 km transect in Scotland with a large urbanization gradient. Supplementary foodstuffs were present in the majority of samples, with peanut (*Arachis hypogaea*) the single commonest (either natural or supplementary) dietary item. Consumption rates exhibited a distance decay from human habitation but remained high at several hundred metres from the nearest household and continued to our study limit of 1.4 km distant. Supplementary food consumption was associated with a near quadrupling of blue tit breeding density and a 5-day advancement of breeding phenology. We show that woodland bird species using supplementary food have increasing UK population trends, while species that do not, and/or are outcompeted by blue tits, are likely to be declining. We suggest that the impacts of supplementary feeding are larger and more spatially extensive than currently appreciated and could be disrupting population and ecosystem dynamics.

## 1. Introduction

Supplementary feeding of garden wildlife is the most common active form of human–wildlife interaction and occurs globally [1,2]. It is particularly widespread in the Western world, with over half of all households participating in many northern European and North American countries, providing an ever-increasing variety and abundance of foodstuffs and feeder designs targeting ever-more diverse species each year [3,4]. Supplementary feeding regimes have also changed in the recent past, with a switch to year-round instead of winter-only feeding. Many mammal and insect species are intentionally provided with supplementary food, but bird feeding is the commonest activity [2,5]. In the UK, for example, the wild bird food market is estimated to be worth £241 million and supply around 150 000 tonnes of supplementary food annually [6], while in the USA over 500 000 tonnes are supplied annually [2,7]. In the UK, there is estimated to be one supplementary bird feeder per nine feeder-using birds [5], providing enough resources nationally to feed three times the entire breeding populations of the ten commonest feeder-using species year-round if they consumed nothing else [8]. Many mammal species such as squirrels and rats also use these resources incidentally but at high frequencies [2,9,10]. While garden wildlife feeding is actively and enthusiastically encouraged by conservation organizations in a majority of countries, including the UK and USA [11,12], such an enormous resource addition is likely to have profound effects on both the organisms benefitting

from it and their natural competitors and prey, and these effects are far from well understood [1,3,13,14].

To date, research into the direct effects of supplementary garden wildlife feeding on the species using it has developed a rather contradictory and mixed evidence base. While some studies have found that supplementary feeding advances breeding phenology and improves reproductive success due to increased resources [15–17], others have found the opposite, possibly due to poor nutrition [18,19]. Similarly, some studies have found benefits to recipient health [20] while others have found detrimental effects [21]. Population- or species-level health is also at risk as promoting artificial long-term aggregations of novel individual and species interactions has facilitated disease spread and crossover, causing large declines in some susceptible species [22,23]. There is consensus as to overwinter survival benefits, to such an extent that migration patterns can be altered due to novel year-round resources [24,25]. Such a large-scale change in diet and feeding behaviour is also likely to have further effects that are just being realized, such as changes to blood chemistry [26] and evolutionary traits [27]. Elucidating any effects on the breeding ecology of feeder-using species is particularly important due to the immediate fitness and population impacts.

One reason why the evidence is conflicting may lie in many studies not being able to account accurately for supplementary food uptake rates in their study organisms due to difficulties in diet detection, and without this critical information it is impossible to assess large-scale impacts and background consumption rates [1,13]. The advent of faecal metabarcoding provides a mechanism whereby this can in part be overcome [28]. This method detects fragments of prey DNA contained within faeces non-invasively, and while the technique is in its infancy and primarily applied to insect prey DNA [29,30], many food types can be distinguished, including plants [31] which are traditionally the commonest supplementary foods provisioned for garden wildlife [1,8].

Most studies to date have provided additional experimental supplementary food and assumed a distance decay in uptake [18,19], however this has two major caveats. First, it does not account for background supplementary feeding rates from resources provided by the local human population unconnected to the study itself, with such cross-contamination rates probably high due to the ubiquity of supplementary provisioning [1,5]. Second, the distance decay rate is unknown in most species and therefore supplementary feeding may be impacting over a greater spatial scale than imagined [17,32]. Assessing diet composition directly through faecal metabarcoding without providing additional experimental resources addresses these caveats.

The largely unknown distance over which the impacts of supplementary feeding occur is also evidenced by the majority of research explicitly stating that impacts are only, or overwhelmingly, encountered in the urban environment, even altering community structure there [33–35], overlooking the wider rural environment. This ignores the ability of many provisioned bird, mammal and insect species to move large distances in search of reliable feeding opportunities [32], and that rural human dwellings are likely to provide more supplementary food per household than urban dwellings [4]. Additionally, research has focused solely on the species that commonly use supplementary feeding without considering those that do not, or do so to a lesser extent. It is likely that increases in populations of common supplementary-food-using species [33] and individual competitiveness will have a negative effect on their competitors and prey species that do not benefit, or benefit less, from supplementary feeding, as background habitat availability is unchanged. Furthermore, if the effects of supplementary feeding are felt over a wider area than solely urban environments, the impacts on community composition and conservation could be far-reaching [16]. Therefore, it is crucial to understand over what distances feeder-using species are travelling to make use of supplementary food resources and what impact this is having upon their ecology, fitness and populations.

In this study, we analyse data from a widespread and common European avian supplementary food user, the blue tit (*Cyanistes caeruleus*), across a 220 km transect of Scotland [36] with a large gradient in distance to human habitation and therefore access to supplementary feeding. We use faecal metabarcoding to uncover what proportion of faeces contains supplementary garden bird food immediately prior to breeding and over what distance supplementary food use is occurring, predicting that use will decline with increasing distance. We then use site average supplementary food use to determine effects on breeding ecology. Finally, we use long-term UK-wide survey data to address the broader implications by assessing whether the utilization of supplementary food is affecting recent population trends in blue tits and their competitors (insectivorous forest bird species) across the UK, hypothesizing that if supplementary feeding is supporting higher populations of recipient species, these inflated populations may be having detrimental effects on the populations of competitor species, contributing to human-mediated homogenizing impacts on biodiversity [37,38]. We believe that this focal study system is highly representative of many supplementary feeding systems and that insights garnered should extrapolate across many systems.

## 2. Methods

### (a) Field data collection
Field data were collected from a 220 km transect of 39 predominantly deciduous Scottish woodland study sites during the springs of 2014–2016 [36]. Study sites were distributed roughly evenly along an approximate south-north line from Edinburgh (55°98′ N, 3°40′ W) to Dornoch (57°89′ N, 4°08′ W) (electronic supplementary material, figure S1). At each site, there were six Schwegler 1B 26 mm hole nest-boxes distributed at approximately 40 m intervals. From mid-March in both 2014 and 2015, the base of each nest-box was lined with greaseproof paper which was replaced when damaged or heavily soiled and removed at the onset of nest building or once a bird had attempted removal by pulling it through the hole [29]. Each nest-box was visited on alternate days and all faeces on the greaseproof paper were removed with sterilized tweezers, with up to a maximum of three faeces collected into a 2 ml Eppendorf tube containing pure ethanol, and the number of faeces collected recorded (with the exception of samples in early 2014). Faecal samples were stored at −18°C within a day of collection and transferred to a −20°C freezer at the end of each field season. Samples were collected from 19 March in 2014 and 18 March in 2015 until nest building, giving a median sampling range of 20 days per site in 2014 and 24 days in 2015, and a maximum

sampling range at a site of 34 days. Faecal samples were not collected in 2016.

The date of first egg laying was recorded for each nest-box (taken as the previous day if two eggs found, as blue tits lay one egg daily [39]) and nest-boxes were designated as occupied in a particular year if at least one egg was laid in a nest by a blue tit. Clutch size was counted once all eggs were laid and incubation had begun. All nestlings were fitted with unique alphanumeric metal rings issued by the BTO bird ringing scheme under licence and productivity was defined as the number of nestlings successfully fledged (number of nestlings alive at day 12 after hatching minus number of dead nestlings found in nest-box post-fledging). Parent birds of both sexes were also captured and uniquely ringed under BTO licence, and their mass, sex and age (first year breeder or second year plus) recorded. Latitude (site range 55.98–57.88° N) and elevation (10–433 m) were obtained for each nest-box [36] and the Euclidian distance to nearest human habitation (33–1384 m) was calculated for each nest-box after finding the coordinates of the nearest human dwelling via Google maps [40]. Owing to the high incidence of supplementary bird feeding in the UK [2,4] this should provide a good predictor of feeder availability. However, we note this provides only the lower limit to the potential distance moved so it is a conservative estimate, as visual inspections for bird feeders around each study site revealed no obvious closer anthropogenic food subsidies.

## (b) Molecular protocol and bioinformatics

Of the total 959 faecal samples collected, 793 were used for metabarcoding, selected by balancing subsampling across nest-boxes and dates and enforcing an upper limit of 10 samples per nest-box per year [29]. If multiple faeces were present in the sample tube, part of each was used for DNA extraction. Thirty samples were also processed in duplicate by dividing the faecal sample into two to assess repeatability. Twenty-four experimental controls were also included (six extraction negatives, nine PCR-negatives and nine PCR-positives using *Inga pezizifera* as a non-native plant PCR-positive, with one PCR-negative and one PCR-positive per 96-well plate).

DNA extraction was performed using the QIAamp DNA Stool Mini kit, following a modified protocol that improved yields [29]. PCR amplification of three loci (COI, 16S and rbcL) was performed for the broader project; of particular importance to this study was the rbcL 'minibarcode' designed to detect 184 base pairs of plant DNA. A second PCR subsequently added indexed Illumina adaptors to the amplicons from each sample; amplicons were then multiplexed in three pools and each pool sequenced on an Illumina MiSeq using 150 base pair paired-end reads.

Sequencing reads were demultiplexed into sample-specific sets using the indices added during PCR amplification, trimmed and clustered into molecular operational taxonomic units (MOTUs) as per the bioinformatics protocol detailed in [29] using the UPARSE pipeline with an identity cut-off of 98% [41]. The taxonomic identity of MOTUs was determined using a BLAST search of the reference set of MOTU sequences against the GenBank and BOLD public databases.

Samples were initially screened for the presence of blue tit sequence at the 16S locus and those with fewer than 100 reads of blue tit were excluded from further analyses ($n = 9$) following [29]. No non-blue tit avian DNA was found in any sample. All nine PCR-positive control samples contained MOTUs attributable to *Inga pezizifera* (range of reads = 4007–12 697) and no more than 19 reads of another MOTU. All nine PCR-negative control samples and three of the six extraction negative control samples contained no more than 22 reads of any MOTU. The remaining three extraction negative control samples showed

high numbers of reads ($n = 991$–6302) from contamination by tomato DNA (which would not impact this study), but not by anything else. Systematic contamination at the rbcL locus was investigated by assessing row and column MOTU correlations [29] but no systematic contamination was found. As there were few cases where a control had greater than 20 reads for any non-target MOTU, we adopted 20 reads as the cut-off for classifying a MOTU as being present in a sample.

MOTUs with less than 90% match to their best BLAST hit were then discarded as inconclusively identified. Remaining MOTUs were amalgamated based on their genus-level identification, as identification to species was seen to be unreliable, consistent with previous assessment of this section of rbcL [42]. A total of 185 plant genera were identified and compared with common supplementary garden bird foods to extract relevant genera. All further analyses were carried out only on these identified supplementary food taxa within focal samples (excluding experimental replicate and control samples, and those not confirmed to be from blue tit). Although the detection of plant DNA could be due either to direct ingestion of that plant taxon by a blue tit itself or that plant being secondarily present in the gut of ingested invertebrate prey, our focus on supplementary food taxa not present in the general Scottish environment means our inference almost certainly reflects solely the direct diet of blue tits and not those of their animal prey.

## (c) Statistical analyses

All statistical analyses were conducted in R v. 4.0.2 [43]. The first model examined how supplementary food consumption varied with respect to environmental factors. A binary value of presence/absence of supplementary food in a faecal sample was used as the response variable of a Bayesian generalized linear mixed model (GLMM) in the MCMCglmm package [44], with distance to nearest human habitation, date, elevation, latitude, year and number of faeces in sample as fixed predictor variables. Year and number of faeces (1–3 and unknown) were categorical factors, while the remainder were continuous variables; the latter were mean centred for ease of interpretation [45] and to facilitate model convergence. Distance to human habitation was analysed on the logarithmic scale due to right skewed data and consistent with a decay model. Date was coded as a deviation from the respective sample site mean per year, as different sites and years have different blue tit breeding phenology. Site and nest-box were included as random effects and the model was run for ten million iterations, removing the first 100 000 as burn-in and thinning every 100. A binomial error structure was used along with parameter expanded priors for the variance terms with residual variance fixed at 0.5. Repeatability in the capacity of the molecular methods to detect supplementary food consumption in the faecal samples was analysed by calculating a Jaccard similarity index of the presence of MOTUs for 29 replicate pairs of faecal samples (one was removed during quality control steps above).

The second set of models aimed to infer whether supplementary food consumption affected the breeding parameters and adult condition of blue tits. For this analysis, the mean supplementary food consumption at each site was calculated. Site-level mean consumption was used rather than nest-box level consumption for two reasons: (i) blue tits often do not nest in a nest-box they are roosting in prior to breeding, but rather nearby, precluding direct attribution; and (ii) faeces were only produced in certain nest-boxes so most nesting attempts are not in a nest-box from which faeces were collected. In addition, if we assume that the mean supplementary food consumption at a site is representative of all individuals nesting there then framing the analysis at the site-level benefits the sample size and power. First, nest-box occupancy was treated

as the response variable in a Bayesian GLMM [44] containing mean supplementary food consumption (varying 0–1), elevation, latitude and year (as a factor) as fixed predictor variables, and site and nest-box as random effects, with all numeric predictor variables mean centred. A binomial error structure was used with similar priors to the first model. Similar models were then run with first egg laying date, clutch size, productivity and adult blue tit mass as response variables with Gaussian error structures and no fixed residual variance. In addition to the standard fixed predictor variables mentioned above, the mass model also contained the age and sex of the bird.

To gain an indication of whether supplementary feeder usage and competition with blue tits may be affecting UK woodland bird populations over time, the 25-year population trends of potential competitor forest bird species were analysed. Twenty species were included based on fulfilling all of the following criteria: average body length less than twice a blue tit (less than 24 cm), foraging substantially on foliage-gleaned invertebrates during the breeding season, occupying wooded habitats, and with a substantial enough UK population to have a 25-year BTO BirdTrends population trend estimate [46], derived from BTO breeding bird surveys. Four categorical variables were coded for each species: their population trend ($1 \leq -50\%$, $2 = -11$ to $-50\%$, $3 = -10$ to $+10\%$, $4 = +11$ to $+50\%$, $5 \geq +50\%$ [46]), supplementary garden bird feeder usage ($1 =$ rare or never, $2 =$ frequent (greater than 5% occurrence in 2020 RSPB Big Garden Birdwatch www.rspb.org.uk/get-involved/activities/birdwatch/results, selected due to the large public inclusion, generating data from a wide range of typical gardens), competition status versus blue tit ($1 =$ outcompeted—average lower mass and/or published evidence of outcompetition for food or breeding sites by blue or great tit [46,47]; $2 =$ not outcompeted—all others), and competition type ($1 =$ food, $2 =$ food and breeding site, if nesting in small cavities [48,49]). Species included, along with their associated data, are shown in electronic supplementary material, table S1. Three Welch's two-sample $t$-tests were then conducted to analyse how population trend varied with regard to (i) supplementary feeder usage, (ii) competition status and (iii) competition type.

## 3. Results

Out of 785 faecal samples, 53% ($n = 414$) contained evidence of supplementary food consumption. Five supplementary plant foodstuffs were identified, with peanut (*Arachis*) by far the most common, present in 49% of total samples. Sunflower (*Helianthus*) was also highly prevalent (17%), with maize (*Zea*) (9%), barley (*Hordeum*) (5%) and millet (*Panicum*) (1%) all rarer. Sixty-three per cent of samples containing supplementary food contained only one type, with 37% containing more than one supplementary foodstuff, and two samples containing all five. Experimental repeatability was high both for detecting peanuts (Jaccard similarity = 0.923) or any supplementary food (Jaccard similarity = 0.923) between replicate samples.

Increasing distance to nearest human habitation predicted a significant reduction in supplementary food consumption (table 1 and figure 1*a*). Different years also exhibited different supplementary food consumption rates. In 2014, the model predicted a 93% chance of a faecal sample containing supplementary foodstuffs at the shortest distances examined in our study (33 m) reducing to 29% at 200 m, 6% at 500 m, and 1% at the furthest site distances examined (1384 m), while in 2015 these figures were higher, with 97% chance at 33 m, 51% chance at 200 m, 15% chance at 500 m and 2%

**Table 1.** Environmental predictors of the consumption of supplementary food by blue tits. Results are taken from a Bayesian GLMM with categorical error structure and logit link function, showing slope estimates and credible intervals for each fixed and random term, with significance asterisks for significant and near-significant terms (pMCMC $\leq 0.1°$ $\leq 0.05*$ $\leq 0.01**$ $\leq 0.001***$). Numeric predictor variables are mean centred, distance to habitation is log transformed, and date has been adjusted for phenology by representing days before mean first egg laying at a given site within a given year. The intercept value for year is 2014 and for number of faeces is one.

| fixed effects | coefficient (C.I.'s) |
|---|---|
| intercept | −1.12 (−2.64–0.44) |
| distance to habitation | −1.97 (−3.10––0.80)*** |
| days before laying | 0.04 (0.02–0.07)*** |
| elevation | 0.0005 (−0.0096–0.0104) |
| latitude | −0.81 (−2.73–1.08) |
| year 2015 | 0.95 (−0.12–2.03)° |
| faeces = 2 | −0.07 (−1.21–1.07) |
| faeces = 3 | 1.37 (0.42–2.36)** |
| faeces = unknown | 2.25 (0.88–3.67)*** |
| **random effects** | |
| site | 5.85 (1.40–11.44) |
| nest-box | 5.01 (2.27–8.24) |

chance at 1384 m to the nearest human habitation (figure 1*a*). The faeces collected at the site furthest from human habitation did however show supplementary food consumption in 75% of samples in 2015 (figure 1*a*), implying that blue tits are able to commute distances up to and beyond our study limit in order to access supplementary food.

Supplementary food use also significantly declined through the sampling period, in the run-up to breeding (table 1 and figure 1*b*). In 2014, the model predicted a 65% chance of a faecal sample containing supplementary food at the earliest sampling times (70 days before mean first egg laying), declining to 24% by 30 days to egg laying and 7% by egg laying (figure 1*b*). For 2015, these figures were elevated to 83% at the earliest times, 44% in the mid time frame and 17% at egg laying (figure 1*b*). Elevation and latitude showed no significant effect on supplementary food consumption, and combining more faeces per sample increased the likelihood of supplementary food detection (table 1). Site and nest-box random effects explained similar amounts of variance (table 1).

Increased supplementary food consumption significantly predicted a large increase in nest-box occupation, from a 20% likelihood with no supplementary food consumption to a 75% likelihood with supplementary food present in every faecal sample (table 2 and figure 2*a*). Supplementary food consumption also significantly advanced egg laying date by five days (from day 122 to day 117; table 2 and figure 2*b*). However, it did not significantly affect clutch size, productivity or the mass of either male or female parent blue tits (table 2).

The $t$-tests showed that the population trends of feeder-using and non-feeder-using bird species were significantly different ($t = -2.3$, d.f. $= 18.0$, $p = 0.03$; figure 3), with feeder-

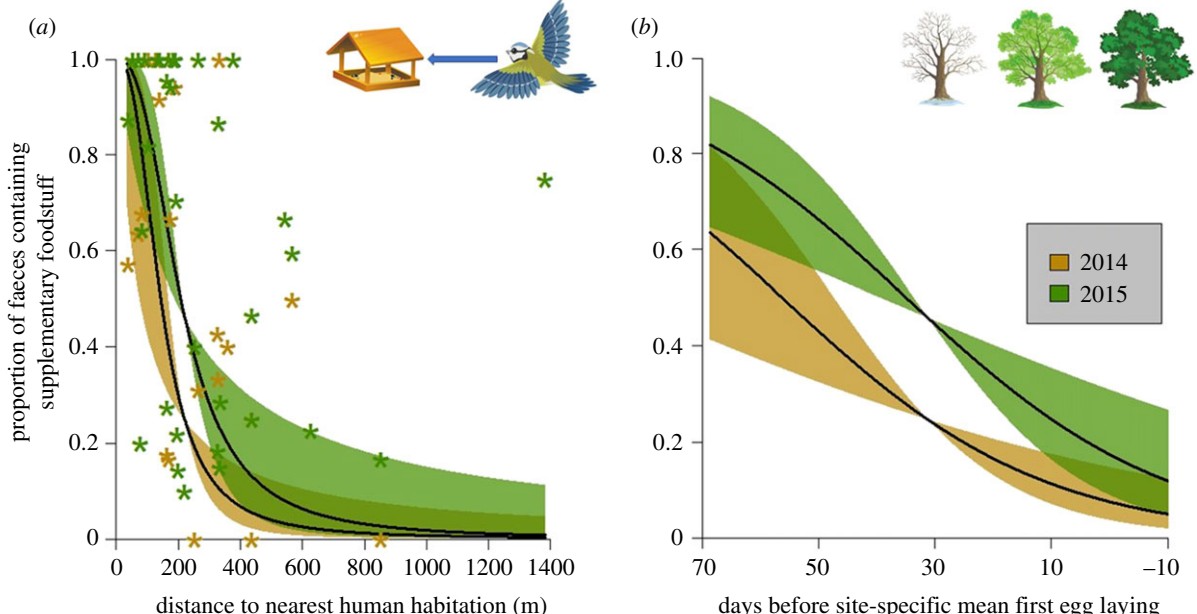

**Figure 1.** Effects of (*a*) distance to nearest human habitation and (*b*) sampling date (adjusted for the phenology of the site within year) on the probability of supplementary food consumption by blue tits. The predicted response of each is shown in 2014 (gold) and 2015 (green). Asterisks in (*a*) show the proportion of faeces containing supplementary foodstuff per site per year (2014 gold, 2015 green), not the per sample presence/absence response analysed in the model. (Online version in colour.)

using species increasing on average and non-feeder-using species decreasing. While competition status ($t = -1.5$, d.f. = 17.8, $p = 0.15$) and competition type ($t = 0.9$, d.f. = 17.7, $p = 0.40$) did not significantly predict population trends, a non-significant suggestion that those species outcompeted by blue tits and competing with blue tits for both food and nesting sites declining more than those not outcompeted by blue tits or only competing for food was observed (figure 3).

## 4. Discussion

This study reveals just how prevalent and ubiquitous supplementary food is in the diet of a wild bird species in a country with high provisioning rates [2,4,32]. Supplementary foodstuffs were shown by faecal metabarcoding to be present in the majority (53%) of blue tit faecal samples immediately prior to breeding, with peanuts identified in more faecal samples (49%) than any other single dietary item, natural or supplementary. For comparison, the most frequent natural prey item, the moth *Argyresthia goedartella*, was present in 34% of the same samples [29]. We show blue tits can travel almost 1.4 km to use supplementary bird feeders during a time of year when movement is thought to be restricted around breeding territories [39]; however, we note our study measured just the distance to the closest human habitation so birds may be moving even further than this. Indeed, as the study area incorporates some of the more remote parts of the UK, these results reveal it likely that supplementary food is available to almost every blue tit in the UK (and other feeder-using bird species, as blue tits are relatively sedentary and short-winged compared to other feeder users [39]), with implications likely to extrapolate across large parts of the Western world due to similarly high supplementary feeding rates [11,12]. We infer from this that any impacts from supplementary feeding will be felt far wider than solely in urban environments as has hitherto been considered [1,33].

As we find that supplementary food usage is strongly associated with a dramatic increase in nest-box occupation (a proxy of breeding density) and an advance in lay date, it is perhaps unsurprising then that we find the national population trends of supplementary feeder-using woodland bird species are increasing on average while the populations of competitor species not benefitting from supplementary feeders are decreasing.

As predicted, supplementary food use declined with increasing distance to nearest human habitation. While this relationship has previously been assumed [18,19], we believe our quantification of it to be the first in a natural situation, made possible by the use of a highly repeatable faecal metabarcoding procedure. Supplementary food usage was still considerable at several hundred metres from the closest potential feeder, yet this distance is greater than the cut-off distance used between treatments and/or nearby human habitation in previous supplementary feeding experiments [19]. Therefore, widespread feeder usage may contribute a background or even confounding effect that was inadequately accounted for in many previous experimental contexts. The distance travelled to supplementary food, and overall usage rates, differed markedly between the 2 years in our study, with 2015 having higher uptake rates than 2014. This may be due to 2015 being considerably colder across our study region, as natural food levels are lower in these conditions [29,50], and benefits from supplementary feeding larger due to natural nutrient limitation, concurring with previous studies [15]. In addition to a distance decay, supplementary food usage also declined over a temporal gradient throughout our study period. This is presumably due to large increases in natural invertebrate prey as spring progresses [29,51] alongside individuals being more restricted to breeding territories as nesting commences [39]. There was no impact of the geographical gradients of latitude and elevation, which vary substantially over the study region, indicating a spatially widespread similarity in feeder usage.

**Table 2.** The effects of supplementary food intake and other environmental variables on a range of blue tit breeding parameters. Results are taken from Bayesian GLMMs, showing slope estimates and credible intervals (CIs) for each fixed and random term, with significance asterisks for significant terms (pMCMC $\leq$0.05* $\leq$0.01** $\leq$0.001***). Occupancy is presented on a binomial scale with the other response variables Gaussian. Numeric predictor variables are mean centred, and the intercept year is 2014, with intercept values of first year adult and female for the mass model. Supplementary food intake was calculated as a per site per year proportion of faeces containing supplementary food which was applied to all nests at that site in that year.

| fixed effects | occupancy coefficient (CIs) | egg laying coefficient (CIs) | clutch size coefficient (CIs) | productivity coefficient (CIs) | mass coefficient (CIs) |
|---|---|---|---|---|---|
| intercept | 0.06 (−0.44–0.60) | 119.33 (117.57–121.06) | 8.70 (8.28–9.12) | 6.96 (6.25–7.63) | 10.71 (10.57–10.84) |
| supplementary food | 2.47 (1.14–3.70)*** | −4.39 (−8.50−−0.17)* | −0.15 (−0.96–0.62) | 0.87 (−0.47–2.11) | 0.07 (−0.12–0.25) |
| elevation | −0.01 (−0.01−−0.01)*** | 0.02 (0.01−0.03)** | 0.001 (−0.002–0.004) | 0.002 (−0.003–0.006) | −0.0006 (−0.0012–0.0002) |
| latitude | −1.30 (−2.11−−0.61)*** | 0.79 (−1.82–3.20) | −0.55 (−0.99−−0.09)* | −0.77 (−1.52−−0.07)* | −0.13 (−0.25−−0.03)* |
| year 2015 | 1.20 (0.61–1.74)*** | 4.41 (2.89–6.11)*** | −1.13 (−1.59−−0.66)*** | −3.41 (−4.17−−2.61)*** | 0.003 (−0.14–0.12) |
| year 2016 | 0.60 (0.08–1.18)* | 8.19 (6.64–9.75)*** | −0.64 (−1.14−−0.18)** | −1.54 (−2.34−−0.74)*** | −0.13 (−0.27−−0.002) |
| 2nd year + | | | | | 0.06 (−0.04–0.16) |
| sex male | | | | | 0.05 (−0.04–0.14) |
| **random effects** | | | | | |
| site | 0.71 (0–1.57) | 12.41 (4.29–22.54) | 0.24 (0–0.58) | 0.60 (0–1.42) | 0.008 (0–0.023) |
| nest-box | 0.74 (0–1.69) | 2.76 (0–7.16) | 0.15 (0–0.47) | 0.15 (0–0.58) | 0.003 (0–0.052) |
| residual | 0.5 | 30.98 (25.00–37.09) | 2.90 (2.41–3.42) | 8.19 (6.93–9.52) | 0.23 (0.20–0.27) |

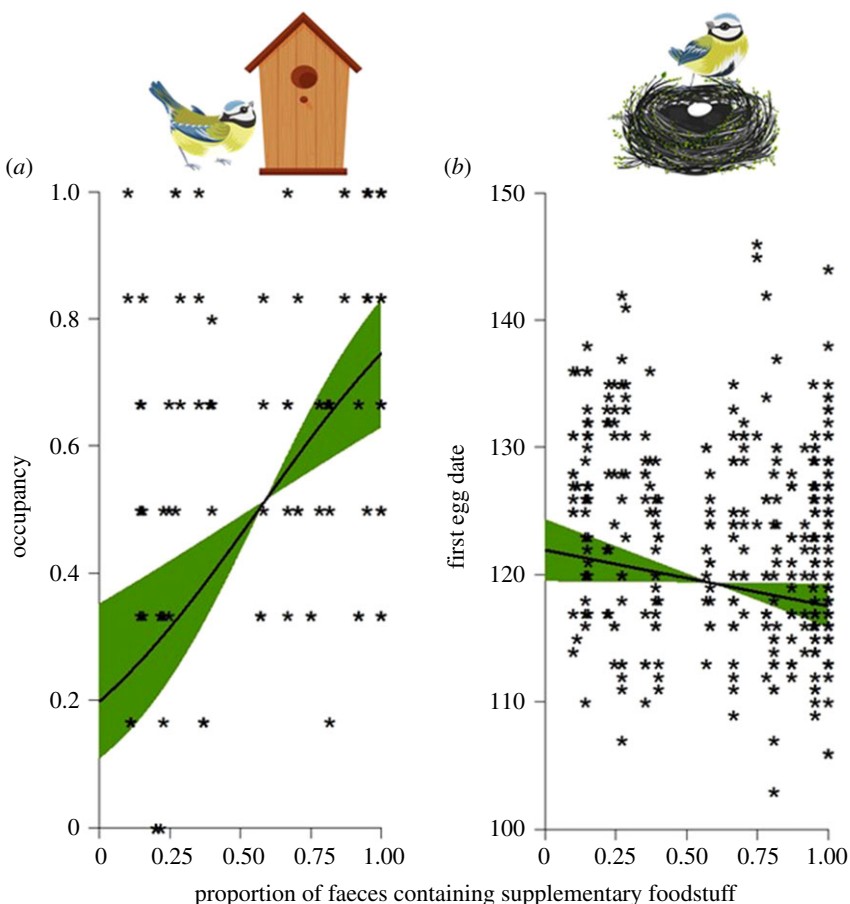

**Figure 2.** Effects of site-level supplementary food intake on (*a*) probability of nest-box occupancy and (*b*) first egg laying date. Asterisks in (*a*) depict site per year occupancy rates rather than the 0/1 occupied response per nest-box analysed in the model. Predictions correspond to 2014. (Online version in colour.)

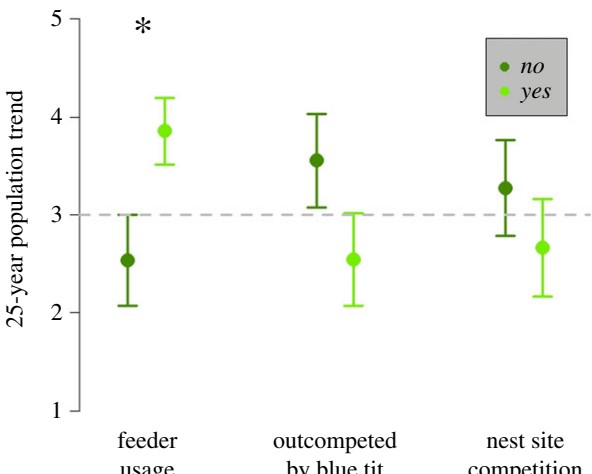

**Figure 3.** Differing population trends of 20 insectivorous woodland bird species in the UK with regard to (i) supplementary garden feeder usage, (ii) behavioural dominance in comparison to blue tits and (iii) whether the species competes for nest sites with blue tits, with values depicting mean ± standard error with an asterisk above a significant difference as determined by a *t*-test. A trend value of 1 represents a large population decline (>−50%) and 5 a large population increase (greater than 50%) over 25 years, with a stable population (trend value = 3) shown by a grey dashed line. (Online version in colour.)

Previous research has developed a mixed picture of the benefits and costs that supplementary feeding confers on the species, including blue tits, using these extra resources [16,21]. Using faecal metabarcoding to identify definite

rather than assumed supplementary food intake without the need for additional experimentation has allowed us to demonstrate major fitness benefits conferred upon blue tits at sites with higher supplementary food uptake. A change in supplementary food use between the observed lowest and highest values predicted an almost fourfold increase in nest-box occupation, an accurate proxy for breeding density in our system due to sites having equal numbers of equally spaced nest-boxes [36]. We expect increasing breeding densities to extrapolate to other species benefitting from feeders as feeder presence increases local abundances of feeder-using species [34], providing an explanation for bird breeding densities covarying with human household densities [52]. However, it is worth mentioning that some infrequent feeder-using species may not experience increased breeding densities due to outcompetition by more dominant beneficiaries. The 5-day advancement in egg laying we identify is very similar to that found in previous food supplementation studies [15,19] and may represent the limit of the plastic phenological response to the lifting of an energetic constraint, with earlier laying associated with higher productivity [53]. Perhaps this mitigation by earlier laying is why individual nest productivity did not decline due to increased density effects associated with feeder usage as might be expected [54], but instead showed a minor increase. Clutch size not being significantly predicted by supplementary feeding agrees with previous studies [19] and reinforces that environmental aspects seem to have little effect on clutch size [36].

Many species using supplementary feeding, such as blue tits, are common, adaptable and already potentially at

population carrying capacity [38,55]. Boosting the productivity, survival, fitness and breeding densities of such species without any increase in available habitat or natural resources is likely to negatively affect their competitors, particularly those not using the new supplementary resources [38]. This may be particularly evident in woodland species compared to farmland species, as rather than replace natural resources that have been lost to all species due to landscape intensification [56], supplementary feeding is providing additional resources solely to certain species. To this end, we demonstrate that populations of UK woodland bird species that don't use supplementary feeders, or only use them infrequently, are likely to be declining, whereas those that use them frequently are likely to be increasing over the last 25 years. Supplementary feeding is therefore probably a driver of population change, in line with other recent evidence [33], however at a much larger scale than solely urbanized environments.

While we do not analyse a causal link between supplementary feeding and the declines of these competitor species, the mechanisms whereby increased blue tit densities could impact other species are clear. For example, blue and great tits are frequently known to evict species such as willow tit and lesser spotted woodpecker from nest holes that they have excavated [47,49], kill pied flycatchers when claiming nesting sites [57], and dominate subordinate marsh and willow tits at food resources [39]. Abundant and permanent feeding might also eliminate any competitive advantage other species (such as marsh tits) exhibit in finding and exploiting natural resources first [58], before being outcompeted by dominant species like blue tits, or migrating to warmer climes for winter to avoid starvation, as for pied flycatchers [59]. Supplementary feeding, therefore, although well-intentioned and beneficial to the species partaking, may be shifting the competitive balance of natural ecosystems and the structures enabling community coexistence, favouring certain species at the expense of others, and contributing to human-mediated ecological homogenization [37,38].

This potential for supplementary feeding to negatively impact bird communities is deserving of further investigation. While we realize that bird feeding will not cease due to its popularity and importance in connecting people to nature and improving human wellbeing [2], it may become necessary for conservation organizations to attempt to limit the impacts. This could include the removal of feeding from nature reserves, a reduction in the encouragement of feeding in areas known to be important for threatened species, or a reversion to winter-only feeding rather than year-round, with each of these potential future research avenues.

In conclusion, we reveal through faecal metabarcoding the pervasiveness of supplementary foodstuffs in the diet of a wild bird and the large benefits using these substantial additional resources confer on its breeding density and phenology. We also show that the distances travelled to use these resources are further than previously imagined, even in a largely sedentary species at a time of year when movement is thought to be restricted. This indicates that the effects of supplementary feeding on ecosystems are likely to extend far beyond just urban environments as has hitherto been considered. Finally, we demonstrate that species making use of supplementary resources are likely to have increasing populations while those that do not are likely to be declining, possibly due to shifting competition balances and ecosystem dynamics. As supplementary provisioning of wildlife (both intentional and incidental) is hugely prevalent and increasing [2], this may have large and widespread ramifications for biodiversity conservation, and we urge caution upon policy makers advocating supplementary feeding for wildlife engagement.

Ethics. Fieldwork was conducted under appropriate British Trust for Ornithology licences (ringing permit A5615 to J.D.S.).

Data accessibility. Data available from the Dryad Digital Repository (https://doi.org/10.5061/dryad.p2ngf1vq2) [60].

Authors' contributions. J.D.S.: Conceptualization, data curation, formal analysis, funding acquisition, investigation, methodology, project administration, visualization, writing-original draft, writing-review and editing; U.H.T.: Data curation; J.A.N.: investigation, methodology and editing.

All authors gave final approval for publication and agreed to be held accountable for the work performed therein.

Competing interests. We declare we have no competing interests.

Funding. This work was funded by an NERC Doctoral Training Studentship (grant no. NE/1338530) to J.D.S. and faecal metabarcoding was funded by NERC (grant no. NE/I020598/1) to Ally Phillimore.

Acknowledgements. We wish to thank Ally Phillimore for commenting on an earlier draft of the manuscript and providing laboratory funding, Jarrod Hadfield for additional laboratory funding, Irene Benedicto Cabello and Ed Ivimey-Cook for assistance with fieldwork, and all the landowners and managers who allowed us access to their land.

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
