## [Peer Review File · Proceedings of the Royal Society B: Biological Sciences]

Review History

RSPB-2021-0480.R0 (Original submission)

Review form: Reviewer 1

Recommendation

Accept with minor revision (please list in comments)

Scientific importance: Is the manuscript an original and important contribution to its field?

Excellent

General interest: Is the paper of sufficient general interest?

Excellent

Quality of the paper: Is the overall quality of the paper suitable?

Good

Is the length of the paper justified?

Yes

Should the paper be seen by a specialist statistical reviewer?

No

Do you have any concerns about statistical analyses in this paper? If so, please specify them explicitly in your report.

No

It is a condition of publication that authors make their supporting data, code and materials available - either as supplementary material or hosted in an external repository. Please rate, if applicable, the supporting data on the following criteria.

Is it accessible?

No

Is it clear?

Yes

Is it adequate?

Yes

Do you have any ethical concerns with this paper?

No

Comments to the Author

This is an important paper, which contributes to broad issues of ecology and conservation biology as well as using recently developed methodology in a new context. It is well written, readily comprehensible and I believe the conclusions follow from the results. I have a few suggestions to improve clarity and I hope also raise the profile of the findings since they are potentially of greater significance than stated here.

1) It would be good to have a map or at least some indicator of where this transect is, other than simply as statement that it is in Scotland.

2) In Fig 1 - there is a notable outlier at 1384m. The text suggests that Blue Tits may fly up to 1400m, but if that outlier is correct, it implies that they might fly considerably further. Whilst the outlying point clearly doesn't distort the trend in the graph, it is a point of considerable interest. therefore we should ask how confident are they in this point? i.e. is it possible that there might have been an anthropogenic food source within 400m, as suggested by the Y axis value? I should like to see some interrogation of that point and a sentence in the results etc about it.

4) Also, it is not clear from the legend of Fig 1 if each point represents a nestbox/bird, or a population mean as it says per site, but is a site a nestbox or a wood? There are more than 39 points on the graph, implying that they are not woods (n=39 in the methods).

5) No reference is made in the text to the data used for the analysis of BTO data, i.e. no reference to supplementary information. I should have preferred the table relating to the analysis of BTO data, which is given in supplementary information labelled populationtrends_dryad to be included as a table in the paper. If left as supplementary, there should be reference to it in the text. I was left wanting to know what were the 21 species used in the analysis and whether I agreed with the assessment of which used bird feeders etc. That analysis, which is presented as a bit of an anticlimax to the paper, is potentially one of the most important findings of the paper, with far-reaching implications.

6) The need for people to connect with nature is immense, both for their own wellbeing and to create a conservation ethos and garden bird-feeding is the most popular and realistic way in which that can be achieved. Garden bird-feeding is not going to go away and for both political and practical and human welfare reasons should not be discouraged. So, armed with that dose of realism, I should like to see some practical suggestions about how these results might be implemented to a) allow bird-feeding and b) reduce the impact of Blue Tits on other species. One

question relates to the provision of nestboxes. There is extensive literature supporting the view that nestboxes encourage Blue Tits (and Great Tits) at the expense of other species. Nestboxes are important ways to engage children in helping nature, as are birdtables. But should we remove all tit nestboxes from our nature reserves? Also, what might be the effect of stopping bird-feeding that favours tits rather than other species say in March or earlier? If the answer is that we need more research, that's fine too, please make that recommendation and highlight the areas that research should focus on, especially e.g. the combined effect of nestboxes and provisioning in or near woodlands. So I'd like to say some translation of these results into potential policy recommendations for the RSPB (not Royal Society Proceedings B!) etc.

7) licence is mis-spelled on lines 139 and 142 - noun licence, verb license.

Review form: Reviewer 2

Recommendation

Accept with minor revision (please list in comments)

Scientific importance: Is the manuscript an original and important contribution to its field?

Excellent

General interest: Is the paper of sufficient general interest?

Excellent

Quality of the paper: Is the overall quality of the paper suitable?

Excellent

Is the length of the paper justified?

Yes

Should the paper be seen by a specialist statistical reviewer?

No

Do you have any concerns about statistical analyses in this paper? If so, please specify them explicitly in your report.

No

It is a condition of publication that authors make their supporting data, code and materials available - either as supplementary material or hosted in an external repository. Please rate, if applicable, the supporting data on the following criteria.

Is it accessible?

Yes

Is it clear?

Yes

Is it adequate?

Yes

Do you have any ethical concerns with this paper?

No

Comments to the Author

Overview

Having recently stumbled across the pre-print for this paper I was please if surprised and slightly amused to be invited to peer-review it. This study addresses an issue that is often glossed over and even potentially confounds quite a lot research on supplementary feeding especially in the UK in terms of its extended area of effect well outside of urban areas. It does this using faecal metabarcoding from samples taken from boxes along transects at the start of the breeding season followed by monitoring of breeding attempts. I believe the use of metabarcoding for this sort of work is both novel and overdue. Further considering the potential effects on other species is a worthwhile inclusion although it is only a first step and so perhaps should not be overplayed too much (which I'm happy to say it hasn't) as it is arguably the weakest part of the paper albeit perhaps the part with the biggest wider implications. All in all, an excellent paper with my recommended changes all being relatively minor.

Abstract

Line 20-21: I'm not sure "assumed" is quite the right word especially given many of the studies looking at things like the effects of breeding performance experimentally (well covered in the references) have taken place in rural(ish) habitat and many (I'd hope all!) researchers in this area are aware that birds are fed outside urban areas where things may be less clear. It's just not been investigated much/at all directly so I think something more along the lines of "not directly considered" or seldom considered/studied etc. I make this point elsewhere too even if it may just be semantics!

Introduction

Line 52: Not vital but I'd also suggest Hanmer et al 2017 (ref 3) here given it deals with supplementary feeding and nest predation.

Line 75: Again not vital but I think giving Orros and Fellowes 2015 (ref 8) would also be a good idea since it gives a relatively up to date overview of food types provided.

Line 86-89: As above more overlook/hard to investigate than not considered.

Line 86-100: Although it is covered to a reasonable extent in the discussion I feel a brief specific mention of species that do to some extent use feeders (even if they are less dominant) but are potentially strongly outcompeted by Blue Tits and other feeder using species eg Great Tit would be appropriate here. These are after all the species that we might think we can help with supplementary feeding but in actual fact we may be harming the most.

Methods

Lines 130-131: More of an observation than a criticism as I'm no expert on DNA extraction related methodology so assume this was all fine but from my limited understanding/experience -20C is lower the temperatures usually used for this sort of thing (-80C?)? Obviously during the field season I understand a domestic freezer is likely to be the only option unless close to base - I assume this was kept to a minimum etc.

Line 139 (and 142): This seems like a very odd way of saying fitted with BTO rings. Please rephrase and mention the scheme ie something like "fitted with unique alphanumeric metal rings issued by the BTO bird ringing scheme under license" - I'm not too bothered about exact wording but I feel it is important to mention the BTO ringing scheme in some form.

Lines 146-149: Was there any potential for additional feeding stations away from gardens? I'd assume not but I have encountered feeding stations in odd places before.

Line 158: Out of curiosity any practical reason for this choice ie was pretty random or even a standard for this sort of thing?

Lines 167-170: I would perhaps prefer a little more detail here even if some of the finer details can be left to the previous paper.

Lines 178-179: The reference to contaminating tomato confuses me...?

Line 200: These may depend on the journal's polices but I would prefer to mention the stats package name in the text (ie MCMCglmm) also that you used R with the standard reference for the version used.

Lines 207-209: Observation - as someone currently learning Bayesian analysis this seems a bit

excessive, but I'll assume it was necessary...

Line 210: Please explain the need to do this a bit more for readers not that familiar with work with DNA extraction data.

Lines 231-246: Strictly speaking I believe these are derived from BTO (and partners) CBC/BBS trends and should be named and referenced as such directly alongside BTO BirdTrends which you've used to make the species selection.

Line 232: Really a point for the discussion but its worth considering how supplementary feeding has changed in that time - Plummer et al 2019 (ref 33) is probably the most relevant paper.

Line 233: Provide a list of species considered, preferably broken down into categories and including references where necessary - supplementary material is probably fine. Also, you've started a sentence with a number which is generally frowned upon...

Line 239-240: Although not a big issue so it doesn't need changing and it probably won't make a real difference it would have been better to use BTO Garden BirdWatch (GBW) survey as a source of data for this sort of thing given the potential issues with the Big Garden Birdwatch dataset.

Line 244: also Parry and Broughton.

Results

General: No real comments - the results are clearly and concisely presented.

Line 252: Observation - I'm quite surprised Sunflower wasn't the highest!

Discussion

Line 310: Again I don't like the word "assumed" in this context.

Lines 336-377: Perhaps reference some of the papers using Blue Tits already covered to drive the point home.

Line 340-343: Perhaps something covered in the previous work on this study system and/or a point for the methods but any feeling for the presence of natural nesting sites in the cavity of the study woodlands? I'd assume its fairly uniform(?) but are they still present or were relatively young woodlands with minimal natural nest sites used? Not really an important issue but it may be useful to help understand breeding density effects.

Lines 343-346: As already mention further consider species that in theory benefit from but also get outcompeted for food and why this might be a conservation issue.

Lines 349-350: I don't entirely understand this point so please explain further.

Lines 354-355: "...already/potentially at carrying capacity..." or similar might be more accurate?

Line 368-379: As I said in my overview this is perhaps the weakest part of the paper but I think this discusses it well. Perhaps add something along the lines of requiring further investigation given the big implications this potentially has for bird feeding as currently carried out in the UK not to mention businesses and charities gaining income from it.

Lines 390-392: Just a suggestion - it may be worth speculating on the potential effects of supplying food over winter but then reducing/stopping a bit before the breeding season itself? Although any such speculation would still need to consider that more Blue Tits are still likely to survive to breed (and compete etc) with winter food provision even if this where to help reduce their effect slightly in the breeding season. I do though wonder if the change from winter only feeding to year round feeding may have had an effect in this area.

Tables and Figures

All very clear and I especially like the little illustrations.

Decision letter (RSPB-2021-0480.R0)

12-Apr-2021

Dear Dr Shutt

I am pleased to inform you that your Review manuscript RSPB-2021-0480 entitled "Faecal metabarcoding reveals pervasive long-distance impacts of garden bird feeding" has been accepted for publication in Proceedings B.

The referee(s) do not recommend any further changes. Therefore, please proof-read your manuscript carefully and upload your final files for publication. Because the schedule for publication is very tight, it is a condition of publication that you submit the revised version of your manuscript within 7 days. If you do not think you will be able to meet this date please let me know immediately.

To upload your manuscript, log into <http://mc.manuscriptcentral.com/prsb> and enter your Author Centre, where you will find your manuscript title listed under "Manuscripts with Decisions." Under "Actions," click on "Create a Revision." Your manuscript number has been appended to denote a revision.

You will be unable to make your revisions on the originally submitted version of the manuscript. Instead, upload a new version through your Author Centre.

1) A text file of the manuscript (doc, txt, rtf or tex), including the references, tables (including captions) and figure captions. Please remove any tracked changes from the text before submission. PDF files are not an accepted format for the "Main Document".

2) A separate electronic file of each figure (tiff, EPS or print-quality PDF preferred). The format should be produced directly from original creation package, or original software format. Please note that PowerPoint files are not accepted.

3) Electronic supplementary material: this should be contained in a separate file from the main text and the file name should contain the author's name and journal name, e.g. `authorname_procb_ESM_figures.pdf`

All supplementary materials accompanying an accepted article will be treated as in their final form. They will be published alongside the paper on the journal website and posted on the online figshare repository. Files on figshare will be made available approximately one week before the accompanying article so that the supplementary material can be attributed a unique DOI. Please see: <https://royalsociety.org/journals/authors/author-guidelines/>

4) Data-Sharing and data citation

It is a condition of publication that data supporting your paper are made available. Data should be made available either in the electronic supplementary material or through an appropriate repository. Details of how to access data should be included in your paper. Please see <https://royalsociety.org/journals/ethics-policies/data-sharing-mining/> for more details.

<http://datadryad.org/submit?journalID=RSPB&manu=RSPB-2021-0480> which will take you to your unique entry in the Dryad repository.

Once again, thank you for submitting your manuscript to Proceedings B and I look forward to receiving your final version. If you have any questions at all, please do not hesitate to get in touch.

Sincerely,
Dr Daniel Costa
mailto:proceedingsb@royalsociety.org

Associate Editor
Comments to Author:

It is unusual to receive feedback from two referees that are so unanimous in their enthusiasm and acknowledgement of the quality of a submitted manuscript. Both reviewers gave high praise to the paper, both in terms of analysis and writing, but also in terms of the conclusions that one can derive from the paper, particularly in terms of ecology and conservation. Importantly, the meaning of supplementary feeding is incredibly topical, as it is at the core of human engagement with nature, a theme of particular importance in a pandemic and post-pandemic society. I highly recommend this manuscript for publication.

Both reviewers have made a series of excellent suggestions - please address them carefully. Please also note that the table of 25 species should be included with references either in the paper itself or perhaps more appropriately as supplementary material.

Marta Szulkin

Reviewer(s)' Comments to Author:

Referee: 1

Comments to the Author(s)

This is an important paper, which contributes to broad issues of ecology and conservation biology as well as using recently developed methodology in a new context. It is well written, readily comprehensible and I believe the conclusions follow from the results. I have a few suggestions to improve clarity and I hope also raise the profile of the findings since they are potentially of greater significance than stated here.

- 1) It would be good to have a map or at least some indicator of where this transect is, other than simply as statement that it is in Scotland.
- 2) In Fig 1 - there is a notable outlier at 1384m. The text suggests that Blue Tits may fly up to 1400m, but if that outlier is correct, it implies that they might fly considerably further. Whilst the outlying point clearly doesn't distort the trend in the graph, it is a point of considerable interest. therefor we should ask how confident are they in this point? i.e. is it possible that there might have been an anthropogenic food source within 400m, as suggested by the Y axis value? I should like to see some interrogation of that point and a sentence in the results etc about it.
- 4) Also, it is not clear from the legend of Fig 1 if each point represents a nestbox/bird, or a population mean as it says per site, but is a site a nestbox or a wood? There are more than 39 points on the graph, implying that they are not woods (n=39 i the methods).
- 5) No reference is made in the text to the data used for the analysis of BTO data, i.e. no reference to supplementary information. I should have preferred the table relating to the analysis of BTO data, which is given in supplementary information labelled populationtrends_dryad to be included as a table in the paper. If left as supplementary, , there should be reference to it in the text. I was left wanting to know what were the 21 species used in the analysis and whether I agreed with the assessment of which used bird feeders etc. That analysis, which is presented as a bit of an anticlimax to the paper, is potentially one of the most important findings of the paper, with far-reaching implications.

6) The need for people to connect with nature is immense, both for their own wellbeing and to create a conservation ethos and garden bird-feeding is the most popular and realistic way in which that can be achieved. Garden bird-feeding is not going to go away and for both political and practical and human welfare reasons should not be discouraged. So, armed with that dose of realism, I should like to see some practical suggestions about how these results might be implemented to a) allow bird-feeding and b) reduce the impact of Blue Tits on other species. One question relates to the provision of nestboxes. There is extensive literature supporting the view that nestboxes encourage Blue Tits (and Great Tits) at the expense of other species. Nestboxes are important ways to engage children in helping nature, as are birdtables. But should we remove all tit nestboxes from our nature reserves? Also, what might be the effect of stopping bird-feeding that favours tits rather than other species say in March or earlier? If the answer is that we need more research, that's fine too, please make that recommendation and highlight the areas that research should focus on, especially e.g. the combined effect of nestboxes and provisioning in or near woodlands. So I'd like to say some translation of these results into potential policy recommendations for the RSPB (not Royal Society Proceedings B!) etc.

7) licence is mis-spelled on lines 139 and 142 - noun licence, verb license.

Referee: 2

Comments to the Author(s)

Overview

Having recently stumbled across the pre-print for this paper I was please if surprised and slightly amused to be invited to peer-review it. This study addresses an issue that is often glossed over and even potentially confounds quite a lot research on supplementary feeding especially in the UK in terms of its extended area of effect well outside of urban areas. It does this using faecal metabarcoding from samples taken from boxes along transects at the start of the breeding season followed by monitoring of breeding attempts. I believe the use of metabarcoding for this sort of work is both novel and overdue. Further considering the potential effects on other species is a worthwhile inclusion although it is only a first step and so perhaps should not be overplayed too much (which I'm happy to say it hasn't) as it is arguably the weakest part of the paper albeit perhaps the part with the biggest wider implications. All in all, an excellent paper with my recommended changes all being relatively minor.

Abstract

Line2 20-21: I'm not sure "assumed" is quite the right word especially given many of the studies looking at things like the effects of breeding performance experimentally (well covered in the references) have taken place in rural(ish) habitat and many (I'd hope all!) researchers in this area are aware that birds are fed outside urban areas where things may be less clear. It's just not been investigated much/at all directly so I think something more along the lines of "not directly considered" or seldom considered/studied etc. I make this point elsewhere too even if it may just be semantics!

Introduction

Line 52: Not vital but I'd also suggest Hanmer et al 2017 (ref 3) here given it deals with supplementary feeding and nest predation.

Line 75: Again not vital but I think giving Orros and Fellowes 2015 (ref 8) would also be a good idea since it gives a relatively up to date overview of food types provided.

Line 86-89: As above more overlook/hard to investigate than not considered.

Line 86-100: Although it is covered to a reasonable extent in the discussion I feel a brief specific mention of species that do to some extent use feeders (even if they are less dominant) but are potentially strongly outcompeted by Blue Tits and other feeder using species eg Great Tit would be appropriate here. These are after all the species that we might think we can help with supplementary feeding but in actual fact we may be harming the most.

Methods

Lines 130-131: More of an observation than a criticism as I'm no expert on DNA extraction related methodology so assume this was all fine but from my limited understanding/experience -20C is lower the temperatures usually used for this sort of thing (-80C?)? Obviously during the field season I understand a domestic freezer is likely to be the only option unless close to base - I assume this was kept to a minimum etc.

Line 139 (and 142): This seems like a very odd way of saying fitted with BTO rings. Please rephrase and mention the scheme ie something like "fitted with unique alphanumeric metal rings issued by the BTO bird ringing scheme under license" - I'm not too bothered about exact wording but I feel it is important to mention the BTO ringing scheme in some form.

Lines 146-149: Was there any potential for additional feeding stations away from gardens? I'd assume not but I have encountered feeding stations in odd places before.

Line 158: Out of curiosity any practical reason for this choice ie was pretty random or even a standard for this sort of thing?

Lines 167-170: I would perhaps prefer a little more detail here even if some of the finer details can be left to the previous paper.

Lines 178-179: The reference to contaminating tomato confuses me...?

Line 200: These may depend on the journal's policies but I would prefer to mention the stats package name in the text (ie MCMCglmm) also that you used R with the standard reference for the version used.

Lines 207-209: Observation - as someone currently learning Bayesian analysis this seems a bit excessive, but I'll assume it was necessary...

Line 210: Please explain the need to do this a bit more for readers not that familiar with work with DNA extraction data.

Lines 231-246: Strictly speaking I believe these are derived from BTO (and partners) CBC/BBS trends and should be named and referenced as such directly alongside BTO BirdTrends which you've used to make the species selection.

Line 232: Really a point for the discussion but its worth considering how supplementary feeding has changed in that time - Plummer et al 2019 (ref 33) is probably the most relevant paper.

Line 233: Provide a list of species considered, preferably broken down into categories and including references where necessary - supplementary material is probably fine. Also, you've started a sentence with a number which is generally frowned upon...

Line 239-240: Although not a big issue so it doesn't need changing and it probably won't make a real difference it would have been better to use BTO Garden BirdWatch (GBW) survey as a source of data for this sort of thing given the potential issues with the Big Garden Birdwatch dataset.

Line 244: also Parry and Broughton.

Results

General: No real comments - the results are clearly and concisely presented.

Line 252: Observation - I'm quite surprised Sunflower wasn't the highest!

Discussion

Line 310: Again I don't like the word "assumed" in this context.

Lines 336-377: Perhaps reference some of the papers using Blue Tits already covered to drive the point home.

Line 340-343: Perhaps something covered in the previous work on this study system and/or a point for the methods but any feeling for the presence of natural nesting sites in the cavity of the study woodlands? I'd assume its fairly uniform(?) but are they still present or were relatively young woodlands with minimal natural nest sites used? Not really an important issue but it may be useful to help understand breeding density effects.

Lines 343-346: As already mention further consider species that in theory benefit from but also get outcompeted for food and why this might be a conservation issue.

Lines 349-350: I don't entirely understand this point so please explain further.

Lines 354-355: "...already/potentially at carrying capacity..." or similar might be more accurate?

Line 368-379: As I said in my overview this is perhaps the weakest part of the paper but I think this discusses it well. Perhaps add something along the lines of requiring further investigation

given the big implications this potentially has for bird feeding as currently carried out in the UK not to mention businesses and charities gaining income from it.

Lines 390-392: Just a suggestion - it may be worth speculating on the potential effects of supplying food over winter but then reducing/stopping a bit before the breeding season itself? Although any such speculation would still need to consider that more Blue Tits are still likely to survive to breed (and compete etc) with winter food provision even if this were to help reduce their effect slightly in the breeding season. I do though wonder if the change from winter only feeding to year round feeding may have had an effect in this area.

Tables and Figures

All very clear and I especially like the little illustrations.

Author's Response to Decision Letter for (RSPB-2021-0480.R0)

See Appendix A.

Decision letter (RSPB-2021-0480.R1)

05-May-2021

Dear Dr Shutt

I am pleased to inform you that your manuscript entitled "Faecal metabarcoding reveals pervasive long-distance impacts of garden bird feeding" has been accepted for publication in Proceedings B.

Data Accessibility section

Open Access

Paper charges

Sincerely,

Dr Daniel Costa

Appendix A

Associate Editor

Comments to Author:

It is unusual to receive feedback from two referees that are so unanimous in their enthusiasm and acknowledgement of the quality of a submitted manuscript. Both reviewers gave high praise to the paper, both in terms of analysis and writing, but also in terms of the conclusions that one can derive from the paper, particularly in terms of ecology and conservation. Importantly, the meaning of supplementary feeding is incredibly topical, as it is at the core of human engagement with nature, a theme of particular importance in a pandemic and post-pandemic society. I highly recommend this manuscript for publication.

Both reviewers have made a series of excellent suggestions - please address them carefully. Please also note that the table of 25 species should be included with references either in the paper itself or perhaps more appropriately as supplementary material.

Marta Szulkin

** We would like to thank the associate editor and referees very much for their positive, insightful and constructive comments, and we are extremely glad that this manuscript seems to have been enjoyed. We also wish to thank the associate editor for highlighting the topical nature of this manuscript. We have carefully addressed the suggestions given and feel that they have improved the manuscript, with the table of species now included as supplementary material as suggested. Quoted line numbers refer to the tracked changes version of the revised manuscript submitted.

Reviewer(s)' Comments to Author:

Referee: 1

Comments to the Author(s)

This is an important paper, which contributes to broad issues of ecology and conservation biology as well as using recently developed methodology in a new context. It is well written, readily comprehensible and I believe the conclusions follow from the results. I have a few suggestions to improve clarity and I hope also raise the profile of the findings since they are potentially of greater significance than stated here.

** We thank the referee for the positive feedback and helping us raise the profile of the findings.

1) It would be good to have a map or at least some indicator of where this transect is, other than simply as statement that it is in Scotland.

** A map (Fig S1) has now been included in the supplementary material and referenced in the manuscript, with a better description of the location of the transect also placed in the main text on lines 125-8.

2) In Fig 1 - there is a notable outlier at 1384m. The text suggests that Blue Tits may fly up to 1400m, but if that outlier is correct, it implies that they might fly considerably further. Whilst the outlying point clearly doesn't distort the trend in the graph, it is a point of considerable interest. therefore we should ask how confident are they in this point? i.e. is it possible that there might have been an anthropogenic food source within 400m, as suggested by the Y axis value? I should like to see some interrogation of that point and a sentence in the results etc about it.

** We thank the referee for raising this point and agree that this particular point implies that they might fly considerably further. We are confident in the accuracy of this point, having inspected all of our study sites for other sources of anthropogenic food sources at the time (now mentioned on lines 154-7) and double checked the distance to the nearest human dwelling. While it is possible that we have missed an obscure and very hidden form of supplementation, we think this unlikely given the remote location of this site, and therefore have expanded upon this point as suggested by the referee in the results section on lines 282-5.

4) Also, it is not clear from the legend of Fig 1 if each point represents a nestbox/bird, or a population mean as it says per site, but is a site a nestbox or a wood? There are more than 39 points on the graph, implying that they are not woods (n=39 in the methods).

** We apologise for this lack of clarity – these points represent a population mean per study wood/site per year rather than per bird/nestbox. There are more than 39 points on the graph due to means from the two different years being represented separately, with <39 points per colour-coded year. We have clarified this in the legend to Fig 1.

5) No reference is made in the text to the data used for the analysis of BTO data, i.e. no reference to supplementary information. I should have preferred the table relating to the analysis of BTO data, which is given in supplementary information labelled populationtrends_dryad to be included as a table in the paper. If left as supplementary, there should be reference to it in the text. I was left wanting to know what were the 21 species used in the analysis and whether I agreed with the assessment of which used bird feeders etc. That analysis, which is presented as a bit of an anticlimax to the paper, is potentially one of the most important findings of the paper, with far-reaching implications.

** We agree with the referee that this was an oversight and this table should have been included. We have now included it in supplementary material (rather than solely in Dryad as previously) as Table S1 and have referenced this in the text as suggested.

6) The need for people to connect with nature is immense, both for their own wellbeing and to create a conservation ethos and garden bird-feeding is the most popular and realistic way in which that can be achieved. Garden bird-feeding is not going to go away and for both political and practical and human welfare reasons should not be discouraged. So, armed with that dose of realism, I should like to see some practical suggestions about how these results might be implemented to a) allow bird-feeding and b) reduce the impact of Blue Tits on other species. One question relates to the provision of nestboxes. There is extensive literature supporting the view that nestboxes encourage Blue Tits (and Great Tits) at the expense of other species. Nestboxes are important ways to engage children in helping nature, as are birdtables. But should we remove all tit nestboxes from our nature reserves? Also, what might be the effect of stopping bird-feeding that favours tits rather than other species say in March or earlier? If the answer is that we need more research, that's fine too, please make that recommendation and highlight the areas that research should focus on, especially e.g. the combined effect of nestboxes and provisioning in or near woodlands. So I'd like to say some translation of these results into potential policy recommendations for the RSPB (not Royal Society Proceedings B!) etc.

** We thank the referee for raising this excellent point. We have introduced a new discussion paragraph on lines 405-11 to address the suggestions made to outline potential practical policy implications and future research directions given the large, and potentially very negative, effects of bird feeding we find in this manuscript.

7) licence is mis-spelled on lines 139 and 142 - noun licence, verb license.

**** Thank you for bringing this to our attention, it has been corrected.**

Referee: 2

Comments to the Author(s)

Overview

Having recently stumbled across the pre-print for this paper I was please if surprised and slightly amused to be invited to peer-review it. This study addresses an issue that is often glossed over and even potentially confounds quite a lot research on supplementary feeding especially in the UK in terms of its extended area of effect well outside of urban areas. It does this using faecal metabarcoding from samples taken from boxes along transects at the start of the breeding season followed by monitoring of breeding attempts. I believe the use of metabarcoding for this sort of work is both novel and overdue. Further considering the potential effects on other species is a worthwhile inclusion although it is only a first step and so perhaps should not be overplayed too much (which I'm happy to say it hasn't) as it is arguably the weakest part of the paper albeit perhaps the part with the biggest wider implications. All in all, an excellent paper with my recommended changes all being relatively minor.

**** We thank the referee for the positive feedback and agree that our subsequent analysis on potential effects on other species is just a first step but useful to include and this will hopefully provide a springboard for other studies to focus on this in future.**

Abstract

Line2 20-21: I'm not sure "assumed" is quite the right word especially given many of the studies looking at things like the effects of breeding performance experimentally (well covered in the references) have taken place in rural(ish) habitat and many (I'd hope all!) researchers in this area are aware that birds are fed outside urban areas where things may be less clear. It's just not been investigated much/at all directly so I think something more along the lines of "not directly considered" or seldom considered/studied etc. I make this point elsewhere too even if it may just be semantics!

**** We apologise for our clumsy wording and have amended 'assumed' to 'largely considered' in this sentence as suggested (the precise suggested wording could not be implemented in this sentence due to the context, but has been in later points).**

Introduction

Line 52: Not vital but I'd also suggest Hanmer et al 2017 (ref 3) here given it deals with supplementary feeding and nest predation.

Line 75: Again not vital but I think giving Orros and Fellowes 2015 (ref 8) would also be a good idea since it gives a relatively up to date overview of food types provided.

**** We have included the two suggested references in support of the statements.**

Line 86-89: As above more overlook/hard to investigate than not considered.

**** We have changed 'without considering' to 'overlooking' as suggested.**

Line 86-100: Although it is covered to a reasonable extent in the discussion I feel a brief specific mention of species that do to some extent use feeders (even if they are less dominant) but are

potentially strongly outcompeted by Blue Tits and other feeder using species eg Great Tit would be appropriate here. These are after all the species that we might think we can help with supplementary feeding but in actual fact we may be harming the most.

** We thank the referee for this suggestion and have included a mention of these species as suggested on lines 95-8 in the introduction, and again on lines 366-8 in the discussion.

Methods

Lines 130-131: More of an observation than a criticism as I'm no expert on DNA extraction related methodology so assume this was all fine but from my limited understanding/experience -20C is lower the temperatures usually used for this sort of thing (-80C)? Obviously during the field season I understand a domestic freezer is likely to be the only option unless close to base – I assume this was kept to a minimum etc.

** Some studies do indeed use colder freezers, but a -20C freezer is adequate to preserve the DNA, especially in combination with the storage in ethanol, which desiccates the samples and denatures nuclease enzymes, and is the same protocol that we (and others) have used in previous barcoding work.

Line 139 (and 142): This seems like a very odd way of saying fitted with BTO rings. Please rephrase and mention the scheme ie something like “fitted with unique alphanumeric metal rings issued by the BTO bird ringing scheme under license” - I'm not too bothered about exact wording but I feel it is important to mention the BTO ringing scheme in some form.

** These sentences have been amended as per the referee's suggestion.

Lines 146-149: Was there any potential for additional feeding stations away from gardens? I'd assume not but I have encountered feeding stations in odd places before.

** We agree that we have also encountered feeding stations in odd places! While we cannot be absolutely certain that no obscure feeders have been missed, we did thoroughly inspect our study sites and the surrounds for feeders (a point that we forgot to mention in the text, now included on lines 154-7) and none were found, so we are confident that we have been as accurate as possible in this regard. This is now better covered in the manuscript.

Line 158: Out of curiosity any practical reason for this choice ie was pretty random or even a standard for this sort of thing?

** This choice was made for practicality as it allowed one positive and one negative per 96-well plate of extractions (the unit in which the molecular lab procedures were performed). This has now been explained in the manuscript on line 167.

Lines 167-170: I would perhaps prefer a little more detail here even if some of the finer details can be left to the previous paper.

** We have now included some further detail on this procedure as requested in lines 176-9.

Lines 178-179: The reference to contaminating tomato confuses me...?

** We have reworded this sentence in a way we hope reduces confusion, on lines 188-90.

Line 200: These may depend on the journal's policies but I would prefer to mention the stats package name in the text (ie MCMCglmm) also that you used R with the standard reference for the version used.

** We apologise for this oversight and these details have now been included in the text as suggested on lines 210 and 213.

Lines 207-209: Observation – as someone currently learning Bayesian analysis this seems a bit excessive, but I'll assume it was necessary...

** These procedures are indeed rigorous and allow us to have a high degree of confidence in our model results.

Line 210: Please explain the need to do this a bit more for readers not that familiar with work with DNA extraction data.

** This line has been expanded on as suggested by the referee, now lines 223-5 in the tracked changes version of the revised manuscript.

Lines 231-246: Strictly speaking I believe these are derived from BTO (and partners) CBC/BBS trends and should be named and referenced as such directly alongside BTO BirdTrends which you've used to make the species selection.

** We apologise for having missed these specifics and have now included more on line 251.

Line 232: Really a point for the discussion but its worth considering how supplementary feeding has changed in that time – Plummer et al 2019 (ref 33) is probably the most relevant paper.

** We agree that this point is worth considering and has now been included both in the introduction (lines 41-2) and in the new discussion paragraph on lines 405-11.

Line 233: Provide a list of species considered, preferably broken down into categories and including references where necessary - supplementary material is probably fine. Also, you've started a sentence with a number which is generally frowned upon...

** We apologise for this oversight, and in line with reviewer 1 and the AE also, we have now included a table of the species considered along with the relevant information in the supplementary material as Table S1, referenced in the text, in addition to the previous inclusion in a Dryad data file.

Line 239-240: Although not a big issue so it doesn't need changing and it probably won't make a real difference it would have been better to use BTO Garden BirdWatch (GBW) survey as a source of data for this sort of thing given the potential issues with the Big Garden Birdwatch dataset.

** We agree with the referee that GBW data would have also been a good data source to use, and various data sources were considered, with some potential issues affecting all. Big Garden Birdwatch data were chosen in preference to GBW partly due to the convenient open access of the data and larger public inclusion, generating data from a wider and potentially more typical range of gardens than the more heavily nature-friendly participants of GBW, who would perhaps have a greater number and range of bird feeders than the typical garden (although we realise that this is where some of the potential issues the referee mentions can occur). We also agree with the referee that

both data sources are broadly equivalent, equally useful and would give similar results. We have included a small line on dataset choice justification in the manuscript on lines 254-6.

Line 244: also Parry and Broughton.

** We have included this reference at this line as suggested.

Results

General: No real comments - the results are clearly and concisely presented.

Line 252: Observation – I'm quite surprised Sunflower wasn't the highest!

** We were also surprised that sunflower was so much lower than peanut (though still highly represented). We imagine this could be due to a wider range of the public feeding peanuts.

Discussion

Line 310: Again I don't like the word "assumed" in this context.

** We apologise for the clumsy wording again and have changed it to considered here too.

Lines 336-377: Perhaps reference some of the papers using Blue Tits already covered to drive the point home.

** We have referenced as suggested.

Line 340-343: Perhaps something covered in the previous work on this study system and/or a point for the methods but any feeling for the presence of natural nesting sites in the cavity of the study woodlands? I'd assume its fairly uniform(?) but are they still present or were relatively young woodlands with minimal natural nest sites used? Not really an important issue but it may be useful to help understand breeding density effects.

** Natural nesting sites do occur in the study woodlands, although they were not surveyed and we do not think that there would be a significant difference in their availability between sites. From our experience at the field sites, we believe that the density differences inferred from nestbox occupancy are very believable and correlate well with site-level blue tit activity.

Lines 343-346: As already mention further consider species that in theory benefit from but also get outcompeted for food and why this might be a conservation issue.

** We thank the referee for this suggestion and have done so on lines 366-8, as well as in the introduction on lines 95-8.

Lines 349-350: I don't entirely understand this point so please explain further.

** We apologise that this point was unclear and have provided an extended explanation on lines 371-3.

Lines 354-355: "...already/potentially at carrying capacity..." or similar might be more accurate?

** This line has been amended as per the referee's suggestion, line 378.

Line 368-379: As I said in my overview this is perhaps the weakest part of the paper but I think this discusses it well. Perhaps add something along the lines of requiring further investigation given the big implications this potentially has for bird feeding as currently carried out in the UK not to mention businesses and charities gaining income from it.

** We have included some lines about the requirement for further investigation given the implications on line 405-11 in the new discussion paragraph.

Lines 390-392: Just a suggestion - it may be worth speculating on the potential effects of supplying food over winter but then reducing/stopping a bit before the breeding season itself? Although any such speculation would still need to consider that more Blue Tits are still likely to survive to breed (and compete etc) with winter food provision even if this were to help reduce their effect slightly in the breeding season. I do though wonder if the change from winter only feeding to year round feeding may have had an effect in this area.

** While we agree with the referee that this is an interesting point that may have an effect, we do not feel able to add very much to this speculation in this particular manuscript as our study ran up to the breeding season but not into it and feeding uptake rates declined as the breeding season approached. We therefore don't know whether such a feeding regime change would make any difference, as all the supplementary feeding we observe would occur under both feeding regimes. We have now included a little extra information about how supplementary feeding has changed recently in both the introduction and in the new discussion paragraph as suggested on lines 40-42 and 405-11, expanding as far as we feel able on this subject.

Tables and Figures

All very clear and I especially like the little illustrations.

** We thank the referee for their positive comments.